# Design of a Mixed Reality Application for STEM Distance Education Laboratories

**Michele Gattullo \***, **Enricoandrea Laviola**, **Antonio Boccaccio**, **Alessandro Evangelista**, **Michele Fiorentino**, **Vito Modesto Manghisi** **and Antonio Emmanuele Uva**

Department of Mechanics, Mathematics, and Management, Polytechnic Institute of Bari, Via Orabona, 4, 70125 Bari, Italy; enricoandrea.laviola@poliba.it (E.L.); antonio.boccaccio@poliba.it (A.B.); alessandro.evangelista@poliba.it (A.E.); michele.fiorentino@poliba.it (M.F.); vitomodesto.manghisi@poliba.it (V.M.M.); antonio.uva@poliba.it (A.E.U.)
\* Correspondence: michele.gattullo@poliba.it

**Abstract:** In this work, we propose a Mixed Reality (MR) application to support laboratory lectures in STEM distance education. It was designed following a methodology extendable to diverse STEM laboratory lectures. We formulated this methodology considering the main issues found in the literature that limit MR's use in education. Thus, the main design features of the resulting MR application are students' and teachers' involvement, use of not distracting graphics, integration of traditional didactic material, and easy scalability to new learning activities. In this work, we present how we applied the design methodology and used the framework for the case study of an engineering course to support students in understanding drawings of complex machines without being physically in the laboratory. We finally evaluated the usability and cognitive load of the implemented MR application through two user studies, involving, respectively, 48 and 36 students. The results reveal that the usability of our application is "excellent" (mean SUS score 84.7), and it is not influenced by familiarity with Mixed Reality and distance education tools. Furthermore, the cognitive load is medium (mean NASA TLX score below 29) for all four learning tasks that students can accomplish through the MR application.

**Keywords:** mixed reality; distance education and online learning; augmented and virtual reality; improving classroom teaching; mobile learning

## 1. Introduction

There have been numerous attempts to introduce Mixed Reality (MR) in the last thirty years in various fields, such as industry, education, retail, cultural heritage, and medicine. However, we still see occasional use in everyday life, except for entertainment applications, such as social media and television. Even if MR is now one of the emerging topics [1,2] and both pilot studies [3,4] and prototypes [5,6] were also proposed in the educational field, there are still few real applications in schools and universities. One of the reasons is teachers' resistance to introducing new practices in their consolidated processes, as pointed out in [7,8]. The epidemiological emergency due to SARS-CoV-2 offers unique opportunities to effectively use new technologies to maintain a high level of teaching while avoiding or minimizing the physical contact between humans. With the spread of the epidemiological emergency in March 2020, schools and universities were forced to rapidly redefine consolidated processes towards forms of Distance Education (DE).

DE is not a recent concept and has a history that spans almost two centuries [9]. As defined by Moore et al. [10], DE is a form of instruction that occurs between two parties (a learner and an instructor), is held at different times and places, and uses varying forms of instructional materials. DE experiences have grown exponentially in recent years, thanks to increased internet speed. However, they were mostly limited to university courses [11]. The recommendation of social distancing by the World Health Organization forced the

introduction of DE in every type of institution worldwide without adequate preparation, causing troubles for students, teachers, and families.

Laboratory lectures are the most hindered by DE, as observed in [12]. Thus, research studies for STEM ("Science, Technology, Engineering, and Math") laboratory lectures are needed to propose methodologies and case studies to derive established procedures. Furthermore, the proposal of new learning tools should be less traumatic for teachers, with a little effort needed to author new learning content. Established DE practices in STEM didactics are beneficial for the entire community beyond the health emergency. They make practice activities possible even for institutions with no laboratories or not easily accessible, such as due to ongoing renovation, lack of laboratory staff, and safety reasons. DE allows people to learn without interfering with work or other activities, thus making knowledge more accessible to a broader range of people. There are no limitations due to the lectures' locations, saving time, and traveling costs with positive impacts on pollution.

However, there are also drawbacks with DE besides the evident reduction of students' human interaction with classmates and instructors. Clark [12] revealed that one of the disadvantages of DE is the lack of hands-on practical activities during laboratory lectures involving observing, testing, building, and repairing. This aspect is of crucial importance in STEM didactics. Most of the laboratory lectures involve the observation of physical objects or phenomena and, in some cases, manipulating them. The learning aim in these lectures is to help students understand a phenomenon or experiment and associate laboratory equipment with what is written and represented in the books. Examples are the association between machine components and their technical drawing, between parts of the body and anatomy diagrams, or between rocks and pictures of geological sites.

These learning objectives in STEM didactics can also be met through DE, thanks to the use of MR, intended as the possibility to use both Augmented Reality (AR) and Virtual Reality (VR) in the same app [13]. MR has been proposed in the past for learning in primary [14] and secondary schools [15], as well as in universities [16]. However, the analysis of the related work, presented in the following Section 2, revealed some open issues that still limit the use of MR as a learning tool. In particular, the MR applications usually are designed without teachers' involvement with the consequence that virtual contents are pleasant but may be difficult to develop and may distract students during the learning activities, thus producing a high cognitive load [17]. For this reason, the design of an MR learning tool must also consider teachers' needs and fulfill a compromise between pleasant graphics and the effort required to accomplish learning tasks.

The main contribution of this work is the design and a preliminary evaluation of an MR application framework for STEM DE laboratories through a methodology that considers the open issues previously found in the use of MR in education with the aim of overcoming them. The major issues found, in fact, are the lack of teachers' involvement making it difficult to promote the scalability of the MR application and the lack of guidelines for the inclusion of selective learning content to help students without distracting them. Our design methodology results in an application with basic graphics which integrates the traditional didactic material and is developed considering both students' and teachers' opinions during the design process. In this work, we present how our design methodology was applied for the development of an MR application for the case study of an engineering course. In this case, the learning task for students is the understanding of assembly drawings of complex machines, one of the milestones in an engineering course. Besides the case study, the proposed methodology can be applied to DE laboratories in every STEM discipline, as well as the MR application framework can be easily adapted to other similar disciplines.

An important step in the proposed methodology is a preliminary evaluation of the MR application. In our case study, we decided to test the usability of the MR application and the cognitive load while doing the learning activities. In fact, it is not evident from the literature if an MR application with basic graphics, like the one proposed in this work,

may produce a low cognitive load in the learning activities while keeping good usability. Therefore, in this work, we formulated the following research questions:

1. "Which is the usability of the designed MR application? How is it affected by students' familiarity with MR and DE tools?"
2. "Which is the cognitive load associated with the learning tasks in the MR application?"

We answered these research questions through two user studies with different target students. In the first user study, we selected students with different familiarity with MR and DE tools who had already passed the exam. In the second user study, we recruited students who were currently studying the target STEM discipline and thus were more familiar with the learning tasks proposed in the MR application.

Only after this preliminary evaluation, the MR application could be introduced to all the students in a class for a future evaluation of learning improvements.

## 2. Related Work

With the SARS-CoV-2 pandemic, DE has become the norm, and several national and international academic societies have combined resources to assure that continuing education occurs during this difficult time. As noted by [18], this experience is teaching us that the education system must be renewed better to prepare the current student generation for an unexpected future. Teachers are asked to explore new didactic methods and tools rather than adapt their traditional teaching methods to DE. In previous works, many researchers have already studied the possibility of integrating new technologies into traditional teaching. Ref. [19] provided a comprehensive survey about the use of mobile computing devices in the learning environment to obtain several advantages such as accessing information quickly, a variety of ways to learn and situated learning. As stated by [20], only learning from home by DE without teacher–student interaction is not suitable for the long term. Thus, they encourage the Blended Learning approach to integrate face-to-face instruction with IoT-based "things" added in class to create a smart learning environment. Among these, new technologies such as AR and VR can be used in self-paced learning to motivate students through media interactive materials and ubiquitous learning to foster anywhere learning in the real world.

AR makes the teaching and learning processes simpler, further making them more exciting and motivating [21,22]. This technology allows the association between a virtual 3D representation of laboratory equipment and its 2D real representation on a printed drawing, usually in the form of the augmented book, as experienced in [6,15]. Drawings can also be enriched with every additional information (text, videos, pictures) useful for self-learning, as made by [5,23]. This information can also include results of numeric simulations, such as finite element analyses as made by [24,25]. AR allows students to acquire all this knowledge without physically being in the laboratory. The only tools needed are a smartphone or a tablet and printed drawings used as trackable. The list of advantages reported by [7] is very long, and we report only some of them: increasing of learner's achievement and motivation, increased participation in class, increased interaction between student and course content, the opportunity of safe application of dangerous experiments. However, the use of AR for STEM education is still not broad because there are some issues to overcome. For this purpose, Ibáñez and Delgado-Kloos [17] found that: (i) the AR applications in STEM didactic rarely stimulate senses other than sight; (ii) few studies provided students with assistance in carrying out learning activities, with AR promoting distraction and increasing cognitive load for students. Furthermore, ref. [7] listed, among the challenges in AR-STEM studies, teachers' resistance to the use of AR. This is somewhat justified by the time-consuming content development phase.

VR has been proposed to observe an experiment in a virtual laboratory and a detailed reconstruction of laboratory equipment. Students can also interact in the virtual laboratory replicating the tasks conducted in the experiment, as shown by [26,27]. However, such an immersive VR experience requires headsets that may not be available for all students in DE. Moreover, even if many students indicated that the virtual laboratory was a useful resource

for preparing them for the laboratory sessions, some other traditional resources, such as the laboratory manual, the pre-laboratory exercises, and the textbook, were ranked more highly. This result suggests that some of the material from traditional resources should be incorporated into the Virtual Laboratory to increase its usefulness to students. Furthermore, ref. [27] pointed out that virtual environments are rich in graphics demanding for teachers' authoring and distracting from students' learning goals. Thus, virtual worlds were not adequate as learning environments on their own. Additional information is needed to gather all the knowledge regarding the course.

From the analysis of the literature, it is possible to say that, besides the advantages, there are still open issues for the use of MR in STEM didactics, in particular:

- Teachers are scarcely motivated to the use of MR in STEM education;
- A high effort is required for the development of virtual content;
- MR applications have distracting graphics that further do not assist students in carrying out learning activities, whereas increasing cognitive load;
- Integration of traditional didactic material into MR applications is needed;
- MR applications should stimulate senses other than sight.

In our work, we started from these open issues with the aim to overcome them by proposing the design and a preliminary evaluation of an MR application framework for STEM DE laboratories through the methodology presented in the next section.

## 3. Materials and Methods

### 3.1. Design Methodology of an MR Application for a Target STEM Discipline

The design methodology applied for the introduction of MR as an established learning tool for a target STEM discipline is represented by the workflow in Figure 1. It can be applied to DE laboratories in every STEM discipline. A preliminary step of the methodology is examining the main open issues for the use of MR in STEM DE (step A). This step is accomplished both through interviewing people who are using MR in STEM DE and reviewing the literature on this topic. For this reason, it needs to be updated every time a new MR application is designed. The open issues contribute to the definition of user needs, independent of the target STEM discipline. User needs are the improvements as desired by the user and expected to be answered by the proposed system. In our case study, we derived the user needs from the analysis of state of the art, as described in Section 2.

Other user needs specific to the target discipline can be derived from a preliminary analysis of the target discipline (step B). It can be carried on through focus groups with teachers and through a preliminary survey of students who have previously attended the discipline course. The analysis of students' and teachers' opinions allows us to understand how MR can help acquire knowledge usually acquired through face-to-face laboratory activities. In this way, students' and teachers' opinions would allow, from one side, to include in the application what students need for learning and, from the other side, to motivate teachers towards the use of MR in their teaching methods. This last aspect is crucial and novel in our approach, considering that one of the open issues that limit the spread of MR in STEM didactics is teachers' resistance towards these new technologies.

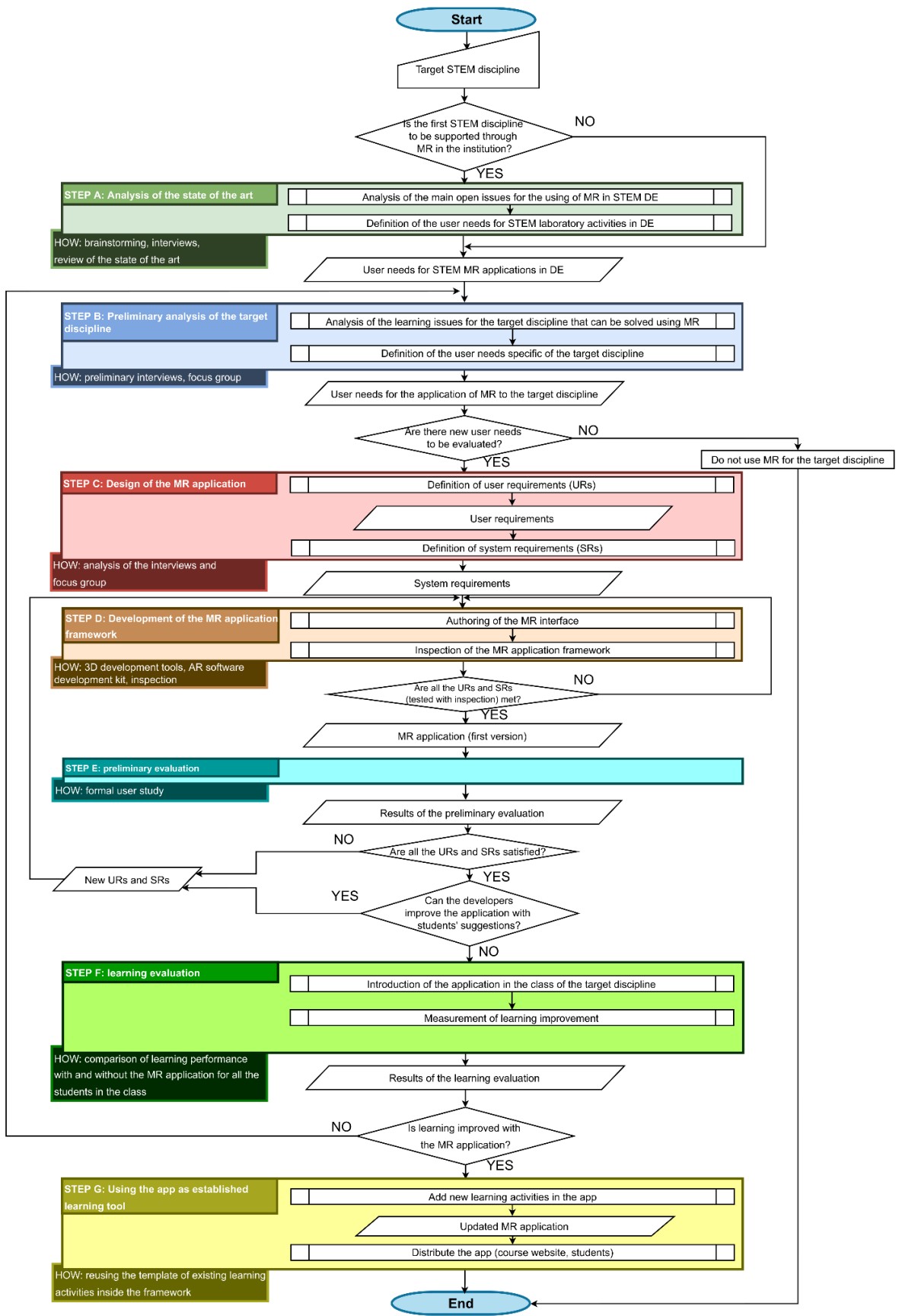

**Figure 1.** Workflow of the design methodology to introduce MR as an established learning tool for a target STEM discipline.

The next step in the design methodology is designing the application by defining the user and system requirements (step C). User requirements (URs) have the goal of checking the functions and the performance that the system will bring to users during its utilization. In our context, they are the functions that the MR application will bring to students and teachers. User requirements are established from the understanding of user needs. They are often written in natural language and do not contain technical details. Every user need can produce one or more user requirements. User requirements are then formalized as system requirements (SRs). They are intended for developers and represent the application designer's statements on what the system must do to meet the user requirements. They contain the technical details needed for the development of the MR application, with a priority defined using the following scale values:

- Must have (M): the requirement must be guaranteed in the system;
- Should have (S): the requirement must be guaranteed but may wait until a second increment;
- Could have (C): the requirement could be implemented, but it is not central to the project objectives;
- Wish to have (W): the requirement will not be implemented, but it will be considered for a future phase.

After preparing the system requirements, it is possible to develop the application framework (step D). During the development, it is possible to verify some system and user requirements through a simple inspection of the MR application. For example, as regards the presence of a needed function. This is an iterative process because the features that do not meet the system and user requirements are redesigned and developed until all the requirements are accomplished. Every development step is then submitted also to the end-users (a small sample of teachers and students), who provide essential feedback for the developers. At the end of this step, the first version of the MR application is released.

The MR application framework is created with the implementation of a few case studies explanatory of the application's final learning aim. Further case studies, which are very important for the effectiveness of a learning tool, could be added once the MR application has been tested. Three-dimensional development tools (e.g., CAD modeling software, Unity 3D, Unreal) are needed to develop the virtual contents and the whole MR experience. Additionally, an AR software development kit, such as Vuforia Engine, is required. The development phase of any mobile application can be facilitated following basic guidelines for the interface authoring. We report some examples from the literature. Ahmad et al. [28] present a literature survey about usability guidelines for mobile applications. These guidelines focus on the visibility of the system status and the consistent representation of information. For a good design of the application, it is necessary to pay attention, for example, to the simplicity of the layout and the logic with which the contents are presented, preferring graphics to texts to reduce as much as possible user's cognitive load. It is also convenient to use quick buttons for frequent operations, borderless buttons, visual alerts, and color contrasts to enhance readability. As to the authoring of virtual contents in an MR application, another literature survey shows how to convey information in Industrial Augmented Reality (IAR) interfaces [29]. As the authors stated, the validity of this study can be extended to other fields, such as the educational one. They classified virtual contents into eight categories: text, signs, photographs, video recordings, drawings, technical drawings, product models, auxiliary models. Therefore, the developer can choose the best virtual content from this list to convey any learning information for the target discipline. A survey with potential developers revealed that the most preferred virtual contents are product models [30].

The next step in the proposed methodology refers to a preliminary evaluation (step E) of the first version of the MR application to test all those user and system requirements that cannot be verified through an inspection (e.g., a measure of the usability) and for which a formal user study is needed. This is another iterative process because if the MR application does not respect these prerequisites, it must be redesigned, modifying the

user and system requirements according to users' feedback received from the preliminary evaluation. Developers can also decide to further improve the MR application by exploiting student feedback, even in the case the preliminary evaluation was successful. In this case, a new preliminary evaluation is needed to verify the improvements of the revised MR application.

The preliminary evaluation is crucial to verify that MR is successfully exploited in the developed application, e.g., without generating a high cognitive load, a bad user experience, and so on. In fact, if the MR application does not satisfy these requirements, students will not be motivated to use it, and then evaluating learning becomes useless. Only once the preliminary evaluation is complete, the MR application can be introduced to all the students in a class, and a second evaluation step will be performed (step F). It aims to verify if the MR application improves student learning performance, comparing a students' sample using the MR application with a sample using traditional didactic material.

If this evaluation is successful, then the application could be used as an established learning tool in the course (step G). As previously said, further learning case studies can be added to the initial framework with a limited authoring effort for the teachers. In fact, these new case studies can be created by copying an existing MR scene in the framework, thus inheriting the same interface and interaction and adapting the contents to the new example. In this way, an updated version of the MR application can always be distributed to students, e.g., on the course website. If the MR application does not show any learning benefit, then some new reasonings are needed about how MR can help students and teachers, defining new user needs. If no new user needs can be defined, the design methodology reveals that MR cannot solve learning issues in the target discipline.

### 3.2. Case Study: Design of the MR Application for the DE Laboratory of "Assembly Drawings of Complex Machines"

The design methodology described in the previous Section was applied to the target discipline of "Methods for Technical Representation" of the bachelor's degree in Mechanics, Management, and Electrical Engineering of the Polytechnic Institute of Bari. In particular, the application helps students in the understanding of assembly drawings of complex machines. Traditionally, this skill is acquired in the first years of the bachelor's degree through laboratory activities where the teacher shows the corresponding components (e.g., bearings, screws, shafts) in a real machine representation in the assembly drawing. These activities are very crucial, as demonstrated by the feedback received from students and teachers in the second semester of 2020 when, during the pandemics, it was not possible to access laboratories.

The analysis of state of the art in this work lets us define four user needs valid for every STEM discipline, reported in Table 1.

**Table 1.** User needs deriving from state of the art.

| ID | User Need | Source | Reference |
|---|---|---|---|
| UN-001 | Students need to be a little distracted by graphics when using the MR application | Students, Teachers | [7,17,27] |
| UN-002 | Teachers need to spend little time in developing virtual contents | Teachers | [7,8] |
| UN-003 | Students need to have the traditional didactic material integrated into the MR app | Students, Teachers | [17,26] |
| UN-004 | Students need to have stimulated senses other than sight | Students | [17] |

Then, we put on a focus group composed of six teachers of the course "Methods for Technical Representation" that listed students' main learning issues that could be solved thanks to MR. Most of them were related to the absence of laboratory activities. They stated that students had difficulty identifying machine components in an assembly drawing, matching the 2D representation of a component and its real shape, and understanding the machine's functionality. The teachers' feedback allowed us to add other user needs to the previous table. They were reported in Appendix A (Table A1).

Furthermore, we interviewed a sample of twenty voluntary students to understand the main issues in learning this subject and suggest improvements to the learning methods. As to the first point, they confirmed the teachers' impressions, while for the second point, they asked for a tool that allowed self-learning, intuitive, that stimulated the study of the subject, and that can be deployed on a mobile phone. Then, we added four other user needs, as detailed in Appendix A (Table A2).

We analyzed the eleven user needs and translated them into nine user requirements listed in Appendix A (Table A3). Finally, we defined seventeen system requirements derived from the user requirements. They, as listed in Table A4 of Appendix A, were used to design the application. Below we describe the initial version of the MR application, resulting from its continuous inspection by the authors involved in the development, the teachers of the course, and a sample of five students.

The application framework was developed with Unity 3D, and the scripts to meet system requirements were written in C#. Vuforia Engine was used for the tracking, i.e., natural feature tracking (Figure 2) was used to recognize the 2D printed assembly drawing, as made in [31]. This solution gives us the possibility to directly associate the drawing of components with the corresponding 3D virtual objects. We experienced a better tracking performance using a larger spacing among the hatch lines in the drawing.

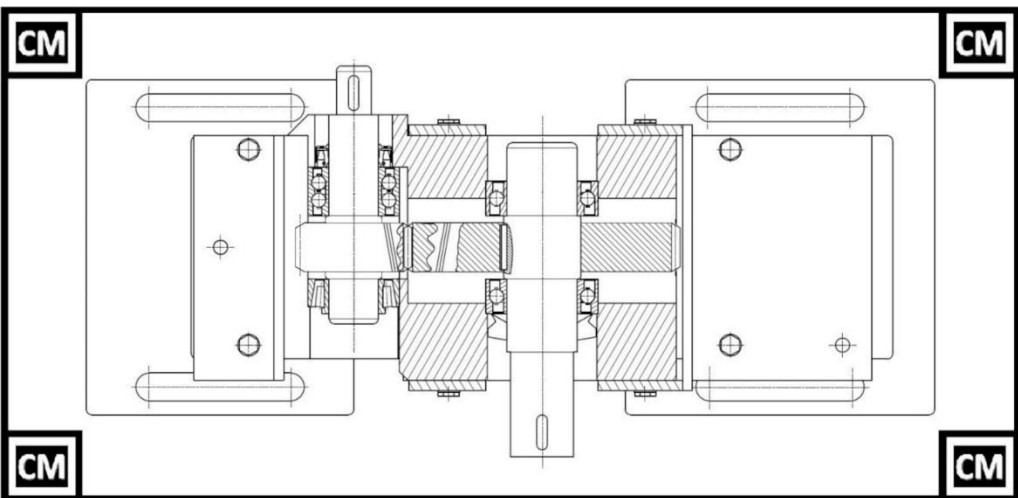

**Figure 2.** Example of assembly drawing used as a natural feature.

The application presents an initial menu (Figure 3) where students can choose the learning activity and the relative drawing to augment. Then, an AR interface appears. In the beginning, no virtual elements are displayed on the drawing, but only GUI (Graphical User Interface) elements: the interactive BOM (Bill of Materials) and the "Highlight", "Balloon", "3D model", "ALL", and "List" buttons. When students select a BOM component, the correspondent portion of the drawing is highlighted (Figure 4). Then, clicking on the other buttons, we verified that the behavior of the app respects system requirements. Only when all the components are highlighted there is the possibility to display the animation of the entire machine, and thus the "Animation" and "Compare" buttons appear (Figure 5).

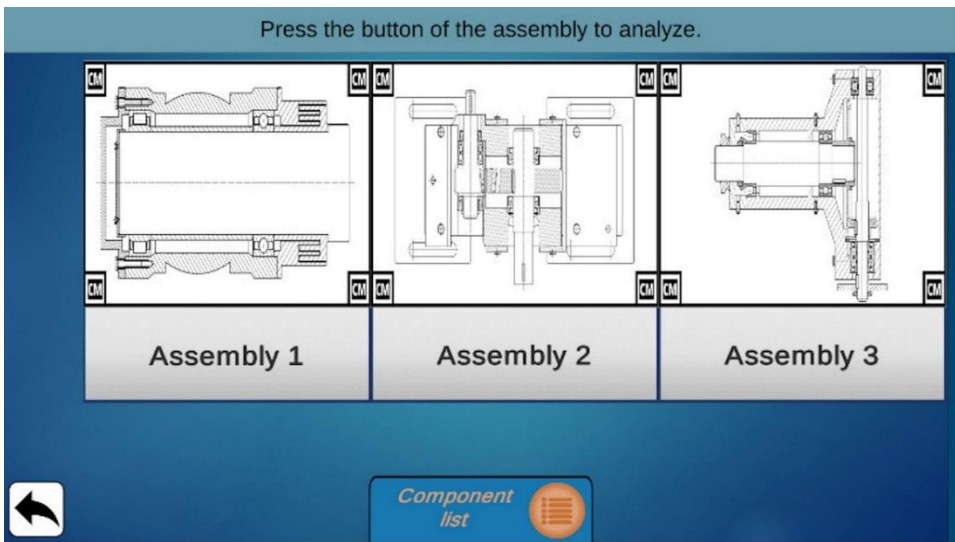

**Figure 3.** Initial menu of the application with the choice of the drawing to be used for the learning activity.

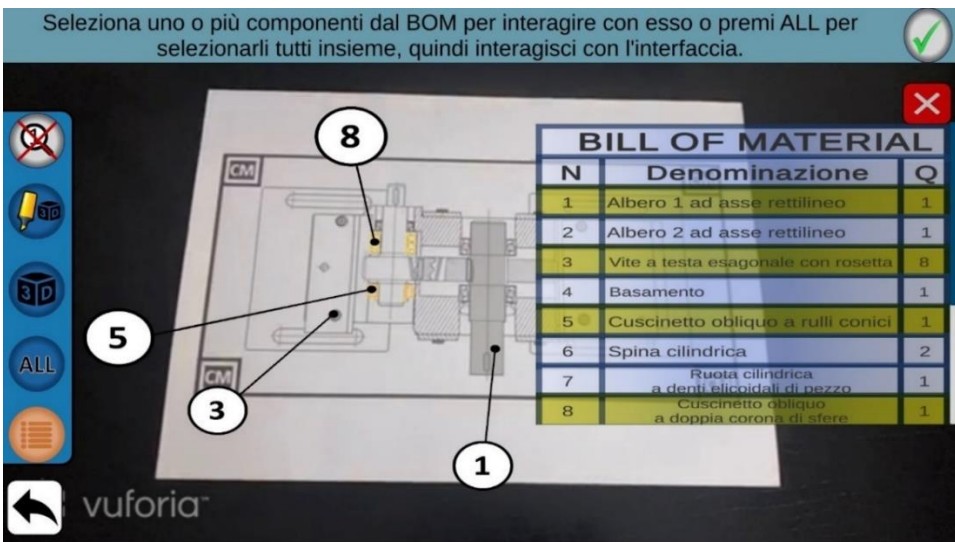

**Figure 4.** AR interface with the interactive BOM (on the right) and the "highlight" and "balloon" visualization enabled for the selected components on the BOM.

The proposed AR interface uses button icons designed to promote "virtual affordance," and significant attention was paid to filtering information, formatting, and relative arrangement of content. These design choices allowed us to have an application easy to use and without distracting elements. All buttons are arranged along the edges of the device screen not to occlude the virtual elements that will be displayed in AR but also to ensure the user the right comfort while pressing them and, at the same time, keeping the display with both hands. Additionally, buttons are positioned on a blue background to ensure a clear reading, thanks to a proper contrast with the real scene.

One of the essential user needs is the integration of traditional didactic material in the application (UN-003). For this user need, we considered that AR is not required. Instead, it could cause distractions and increase the mental effort in using the application. Then, we created a separate VR section. We did not use an immersive VR environment because it would be hard to use on a student's mobile device (UN-011), it would be rich in distracting graphics (UN-001), and the authoring of new content would be more demanding for teachers (UN-002). Therefore, we prefer using desktop VR [32], where users interact with

virtual components using the touch screen of their mobile device. Students can pan, rotate, and zoom the 3D model of a component as conducted in whatever 3D CAD software. This desktop VR section is accessible from both the main menu and the AR GUI, and it allows an in-depth study of all types of components, not limited to those visible in the chosen technical drawings. Components are presented through a menu (Figure 6) with the same classification used in the traditional didactic material. In this menu, buttons are designed to provide the user with a preview of the CAD model that will appear after pressing them. The colors used for the icons and backgrounds are based on different shades of white and blue. After choosing the type of component, the user has access to all the variants of that component (Figure 7). They can be further elaborated through labels, animations, pictures, information filtering, and textual content.

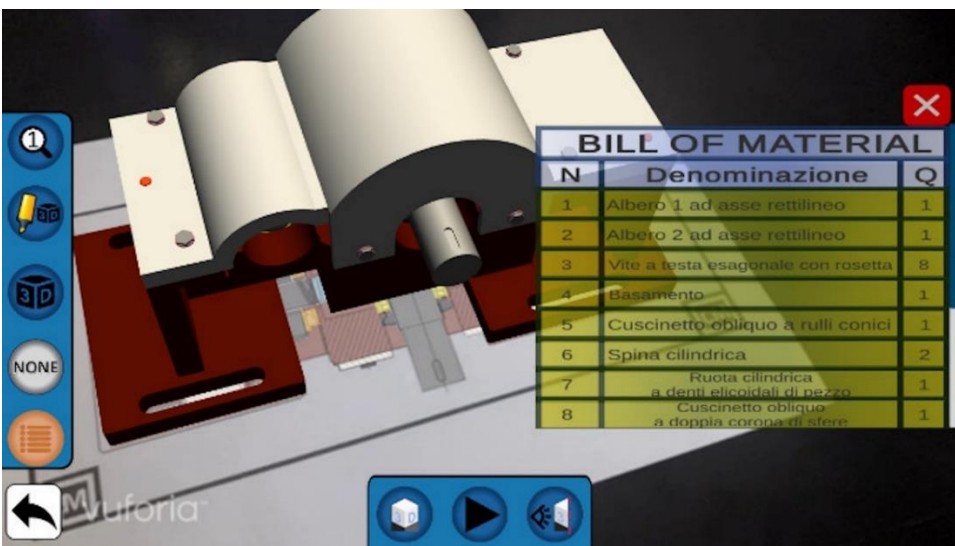

**Figure 5.** AR interface with all the components displayed and the "3D model" visualization enabled. In this visualization mode, there is the possibility of playing an animation or comparing 2D with 3D through the buttons at the bottom of the GUI.

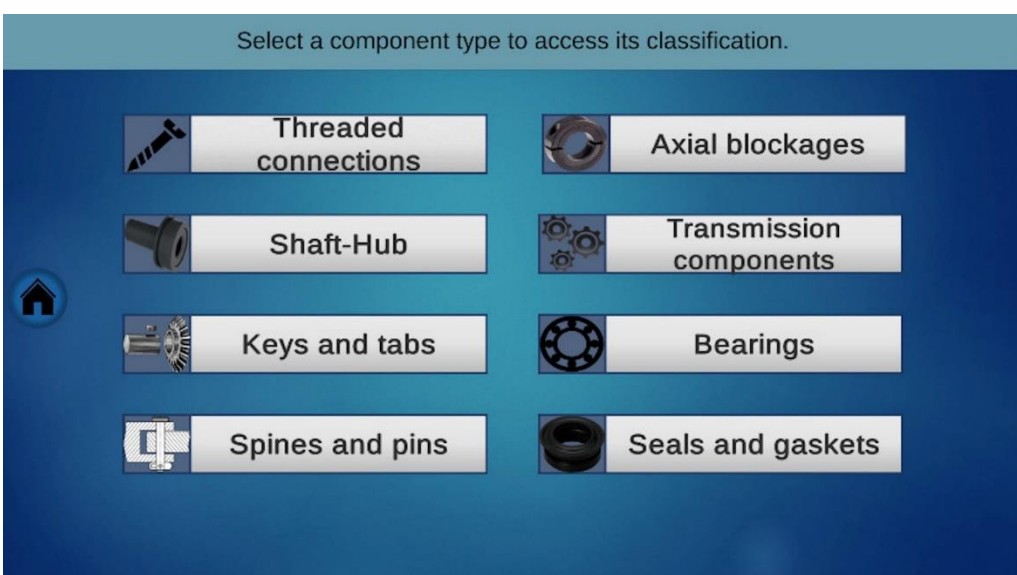

**Figure 6.** Selection menu of the desktop VR section of the application.

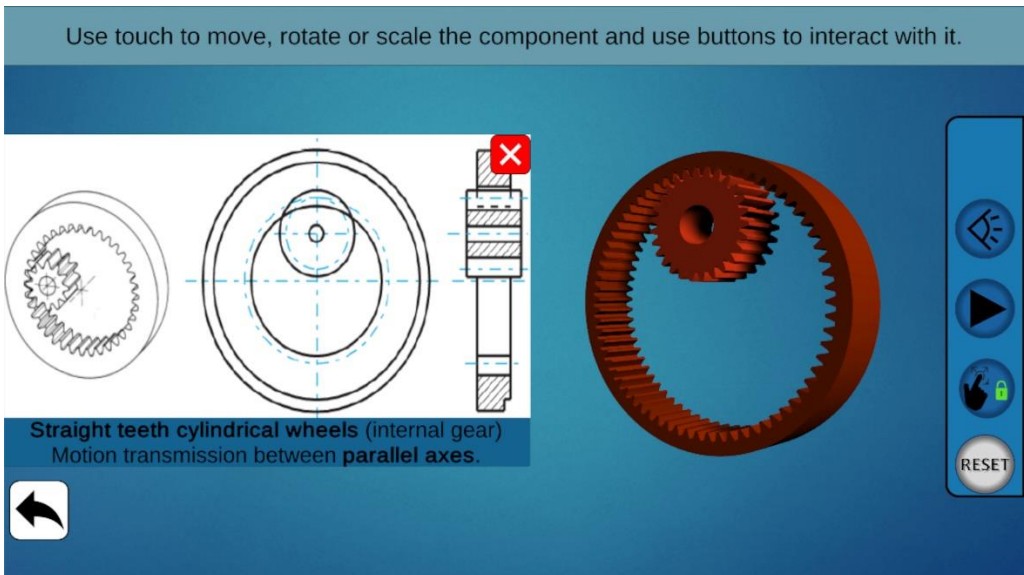

**Figure 7.** GUI elements of the desktop VR section: a navigable 3D model of a component (in the middle), a hidden window with additional pictures with the learning content (**left**), buttons to display an alternative visualization of the component (**right**).

*3.3. Preliminary Evaluation*

The first version of the MR application was used to test UR-001, UR-002, UR-009, SR-015, SR-016, and SR-017, for which formal user studies are needed. This preliminary evaluation is crucial because it is not enough that teachers judge useful a new learning tool. It must also be easy to author with new learning activities. Furthermore, students must also find it easy to use, and the cognitive load associated with the learning tasks must be low; otherwise, they will not use it for their learning, as revealed by the literature [17].

To evaluate the scalability of our framework (UR-009, SR-015), we estimated the time spent to augment a new machine drawing with similar complexity. Figure 8 shows an example of the machine complexity used in the case study with the main components. Then, we showed the application and provided the system requirements to three master's degree engineering students with enough knowledge and experience in CAD modeling and development of MR applications with Unity3D. On average, they spent 27.5 h adding the new drawing and the relative MR content in the application framework (Figure 9). It means that no more than four working days are needed for a skilled student to extend the application with a new drawing/product. This is an interesting result since we spent about fifty working days developing the framework.

To evaluate the usability (UR-001, SR-016) and the cognitive load (UR-002, SR-017), we designed two user studies whose results allowed us to answer the research questions in this work.

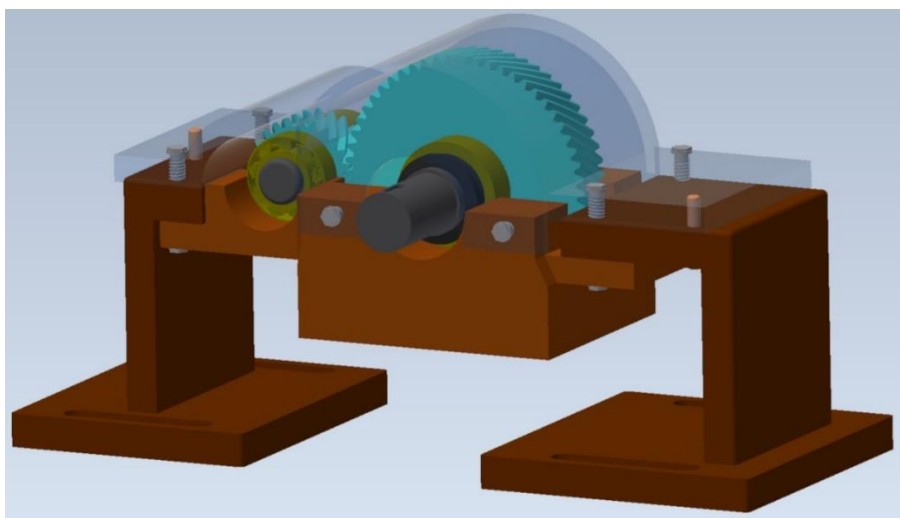

**Figure 8.** Example of the machine complexity used in the case study with the main components.

| N | Activity | Hours needed | Timeline [h] | | | | | | | | | | | | | | | | | | | | | | | | | | | | | | |
|---|---|---|---|---|---|---|---|---|---|---|---|---|---|---|---|---|---|---|---|---|---|---|---|---|---|---|---|---|---|---|---|---|---|---|
| | | | Work day 1 | | | | | | | | Work day 2 | | | | | | | | Work day 3 | | | | | | | | Work day 4 | | | | | | | |
| | | | 1 | 2 | 3 | 4 | 5 | 6 | 7 | 8 | 9 | 10 | 11 | 12 | 13 | 14 | 15 | 16 | 17 | 18 | 19 | 20 | 21 | 22 | 23 | 24 | 25 | 26 | 27 | 28 | 29 | 30 | 31 | 32 |
| 1 | 3D CAD modelling of the assembly | 10 | █ | █ | █ | █ | █ | █ | █ | █ | █ | █ | | | | | | | | | | | | | | | | | | | | | | |
| 2 | Assembly drawing and import in Unity 3D as trackable | 6 | | | | | | | | | | | █ | █ | █ | █ | █ | █ | | | | | | | | | | | | | | | | |
| 3 | 3D CAD model alignment with natural feature | 1 | | | | | | | | | | | | | | | | | █ | | | | | | | | | | | | | | | |
| 4 | Updating BOM and functionalities linked to its components | 2 | | | | | | | | | | | | | | | | | | █ | █ | | | | | | | | | | | | | |
| 5 | Baloons alignment with natural feature | 1 | | | | | | | | | | | | | | | | | | | | █ | | | | | | | | | | | | |
| 6 | Highlight alignment with natural feature | 3 | | | | | | | | | | | | | | | | | | | | | █ | █ | █ | | | | | | | | | |
| 7 | Animations of the assembly | 1 | | | | | | | | | | | | | | | | | | | | | | | | █ | | | | | | | | |
| 8 | Creation of component recognition labels and scripts | 2 | | | | | | | | | | | | | | | | | | | | | | | | | █ | █ | | | | | | |
| 9 | Updating link between AR and VR scenes | 1 | | | | | | | | | | | | | | | | | | | | | | | | | | | █ | | | | | |

**Figure 9.** Gantt chart for the activities needed to add a new drawing and the application's relative virtual content.

### 3.3.1. User Study 1: Usability

The usability can be evaluated through subjective questionnaires such as the System Usability Scale (SUS) questionnaire [33], the questionnaire of User Interface Satisfaction (QUIS) [34], and the Smart Glasses User Satisfaction (SGUS) [35]. Compared to other subjective questionnaires for usability evaluation, we used the SUS because it is the most suitable for our application and was widely used in similar studies before concerning the educational field such as health [36], language [37], chemistry [38], and environmental education [39]. At this point of our research, we asked if the usability evaluation should be conducted by students who had previous experience or not with MR applications and DE tools. Our hypothesis is that students without previous experience in MR could judge more positively the developed MR application because they are more influenced by the wow effect caused by using new technology.

This user study involved 48 students: 13 females, mean age 24.0 (SD 1.4). They were all master's degree students in Mechanical Engineering, except for a Ph.D. student in Mechanical Engineering and a master's degree student in another Engineering faculty. We selected them, ensuring that we had some people with previous experience with MR applications and DE tools. All the students had already passed the "Methods for Technical Representation exam." Five users used the application on a tablet, while the remaining on

a smartphone. The mean time spent getting familiarized with the application was 12.4 min (min 3, max 70, SD 11.5 min). The familiarity levels are shown in Figure 10. From the results of these answers, we divided students into two samples. In the first one (low familiarity), we put all the students that declared to be "not at all familiar"/"slightly familiar" with either one of the three categories. The students in this sample were 28. The remaining 20 students were in the second sample (high familiarity).

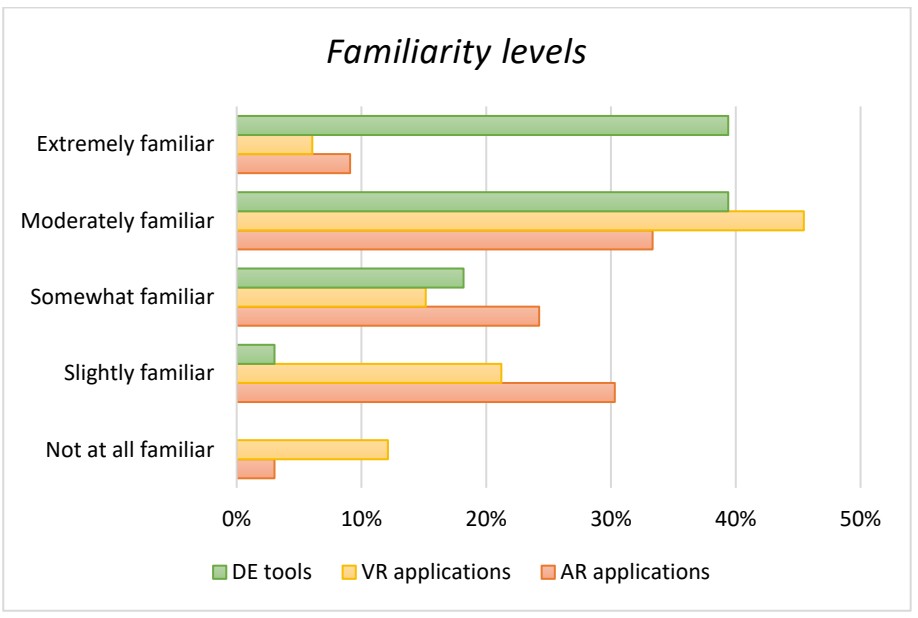

**Figure 10.** Students' familiarity with AR applications, VR applications, and DE tools.

We showed a 1 min video that briefly presents all the functionalities of the application. Then, we asked to download the application and print the markers as preliminary operations. Then, we asked them to familiarize themselves with the application, exploring all the functionalities: localizing components on the drawing, exploring the shape of components, comprehending how the assembly works, and accessing the traditional didactic material. We observed students in this phase and were available to help them in case of difficulties. After these preliminary operations, students compiled the questionnaire. We also asked the level of familiarity with both AR and VR applications, as well as DE tools, using a 5-point Likert scale. After the ten standard queries of the SUS questionnaire, we provided two additional questions aimed at evaluating how the learning could be improved with the MR application, based on students' previous experience with the target discipline. We asked how much students agreed with the following statements, using the same 5-point Likert scale of the SUS. The two statements are (i) "With this application, I can also learn without using the teacher booklets, book, or personal notes", and (ii) "I would prefer to take a closer look at real machine components". Finally, we asked about possible additional functionalities that could be added to the application and general comments, issues, or tips.

### 3.3.2. User Study 2: Cognitive Load

The cognitive load can be evaluated through performance measures, subjective measures (e.g., using the NASA Task Load Index or the Subjective Workload Assessment Technique), or physiological measures [40]. It is important to measure the cognitive load associated with each learning task (e.g., finding an object in a drawing) accomplished through the MR application. Then, this evaluation must be performed with students who are currently studying the target STEM discipline, who know these learning tasks and the workload associated with the traditional didactic material. To evaluate the cognitive load in this work, we could not use the same sample of students who evaluated the usability. The students selected for this evaluation must currently study the target discipline of "Methods

for Technical Representation". They have no experience in MR applications because MR disciplines are taught in the master's degree. Thus, this constraint does not match the one requested for the usability evaluation made by students with previous experience in MR applications. The cognitive load was evaluated using the NASA Task Load Index (TLX) scale, which is considered the most sensitive and reliable subjective measure [41].

As in the previous user study, we showed a 1 min video that briefly presents all the application's functionalities. In the same way, we asked them to download the application, print the markers, and familiarize themselves with the applications, exploring all the functionalities. After these preliminary operations, students were asked to perform four learning tasks:

1. Localize two components on the drawing;
2. Explore the shape of two components;
3. Comprehend how the assembly works;
4. Access to the traditional didactic material for a component.

For each task, detailed textual instruction was provided to the students. They were asked if they were able to accomplish the task "fully", "partially", or they "did not understand what to do". If students provided this last answer, a video showed how to do the task. When students accomplished the task, they answered the NASA-TLX questionnaire.

The survey was provided to 36 students: 7 females, mean age 20.2 (SD 1.4). They were all bachelor's degree students in Electrical Engineering. They were currently attending the discipline of "Methods for Technical Representation". 2 users used the application on a tablet while the remaining on a smartphone.

## 4. Results

### 4.1. User Study 1

We compared the SUS scores of the two samples. Data in the "low familiarity" sample followed a normal distribution (Shapiro–Wilk test: $W = 0.942$, $p = 0.121$), whereas data in the "high familiarity" sample did not follow a normal distribution (Shapiro–Wilk test: $W = 0.864$, $p = 0.009$). Thus, we used the Mann–Whitney U test to compare samples. It revealed no significant difference between the samples ($N = 48$, $U = 222.5$, $p = 0.226$). This result allows us to answer our second research question, saying that the application's usability is not influenced by students' familiarity with AR/VR applications and DE tools. Then, we considered data from all the students as reported in Table S1 of the Supplementary Material, and the mean SUS score was 84.7 (SD 11.0, max 100.0, min 55.0) over a maximum of 100, thus respecting the SR-016. Therefore, using the rating scale provided by [42], we can define the usability of our application as "excellent".

The results of the two additional questions are reported in Figure 11. They revealed that students rated this application complementary but not a substitute for the traditional didactic material. Students always prefer traditional laboratory experience with hands-on real machine components.

As to the students' feedback, we separated comments and issues from additional functionalities and more relevant tips for our study. All comments were favorable judgments on the application: 9 were given from the "high familiarity" group and 13 from the "low familiarity" group. Issues regarded the compatibility with students' devices experienced by four users and tracking lost beyond a certain distance for five users. As expected, students with "high familiarity" provided more recommendations (35 vs. 24). As we can see from Figure 12, we distinguished recommendations into five topics: contents, interaction, GUI, device, and tracking. Detailed answers were reported in Table 2. We analyzed the extensions that we planned to do in the short and mid-term, those that need further evaluation, and those out of the learning scopes.

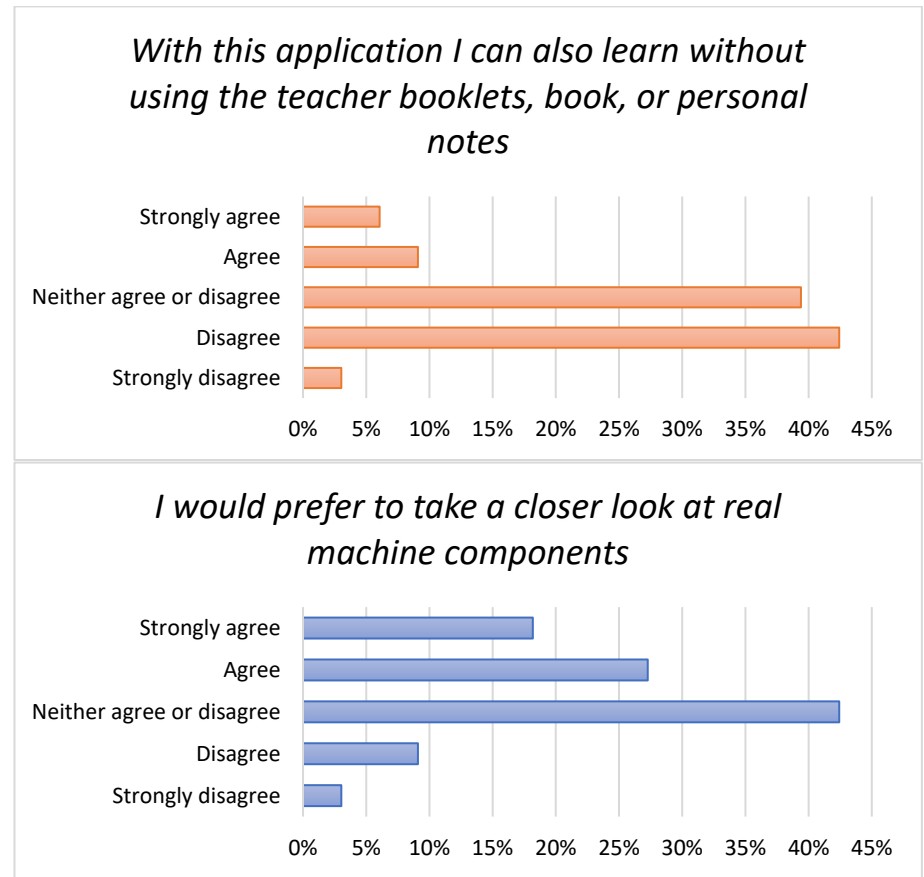

**Figure 11.** Results of the additional questions in user study 1.

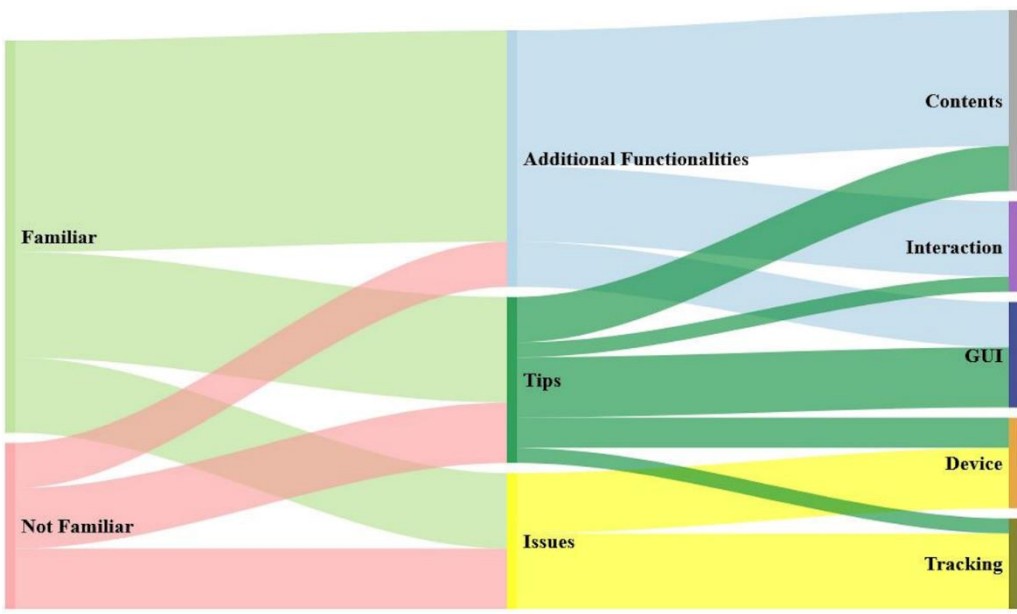

**Figure 12.** Analysis of the recommendations given by the students.

**Table 2.** Recommendations provided by the students.

| Additional Functionality and Tips Suggested by the Students | | | | | | |
|---|---|---|---|---|---|---|
| Recommendation | Topic | N. of Students | Short Term | Mid Term | Further Evaluation Needed | Out of Scope |
| *Simulation of the product kinematics* | Contents | 1 | x | | | |
| *Visualization of the CAD model also for not standard components* | Contents | 1 | | | | x |
| *Technical details (function, dimension, material, permissible load, diagrams)* | Contents | 4 | | | x | |
| *Link for additional information (formula, history, exercises)* | Contents | 2 | | | | x |
| *Assembly placement in a real context* | Contents | 1 | | | x | |
| *Integrated tutorial* | Contents | 2 | x | | | |
| *Audio-description of products* | Contents | 1 | x | | | |
| *Augmentation of more products* | Contents | 1 | | x | | |
| *Extendible application with customized products and markers* | Contents | 1 | | | | x |
| *Make compatible with iOS* | Device | 2 | | x | | |
| *Realistic rendering of CAD models* | GUI | 3 | | | | x |
| *More attractive interface* | GUI | 3 | | | x | |
| *More ergonomic central control bar* | GUI | 1 | x | | | |
| *Zoom lens in the AR scene* | Interaction | 2 | | x | | |
| *Allow users to assemble the product by their own* | Interaction | 1 | | | | x |
| *Save screenshots of AR scene* | Interaction | 1 | x | | | |
| *Auto detection of markers* | Tracking | 1 | | x | | |

*4.2. User Study 2*

We computed the weighted NASA TLX score for each user in each task. Figure 13 shows the scores of the six workload parameters evaluated in the four learning tasks. Then, we evaluated the mean NASA TLX for each task. For task 1 ("localize two components on the drawing"), the mean weighted NASA TLX score was 26.3 (SD 17.3, max 67.3, min 0) over a maximum of 100. For task 2 ("explore the shape of two components"), the mean weighted NASA TLX score was 22.0 (SD 18.6, max 62.3, min 0) over a maximum of 100. For task 3 ("comprehend how the assembly works"), the mean weighted NASA TLX score was 19.9 (SD 18.2, max 64.7, min 0) over a maximum of 100. Finally, for task 4 ("access to the traditional didactic material for a component"), the mean weighted NASA TLX score was 23.3 (SD 20.3, max 67.0, min 0) over a maximum of 100.

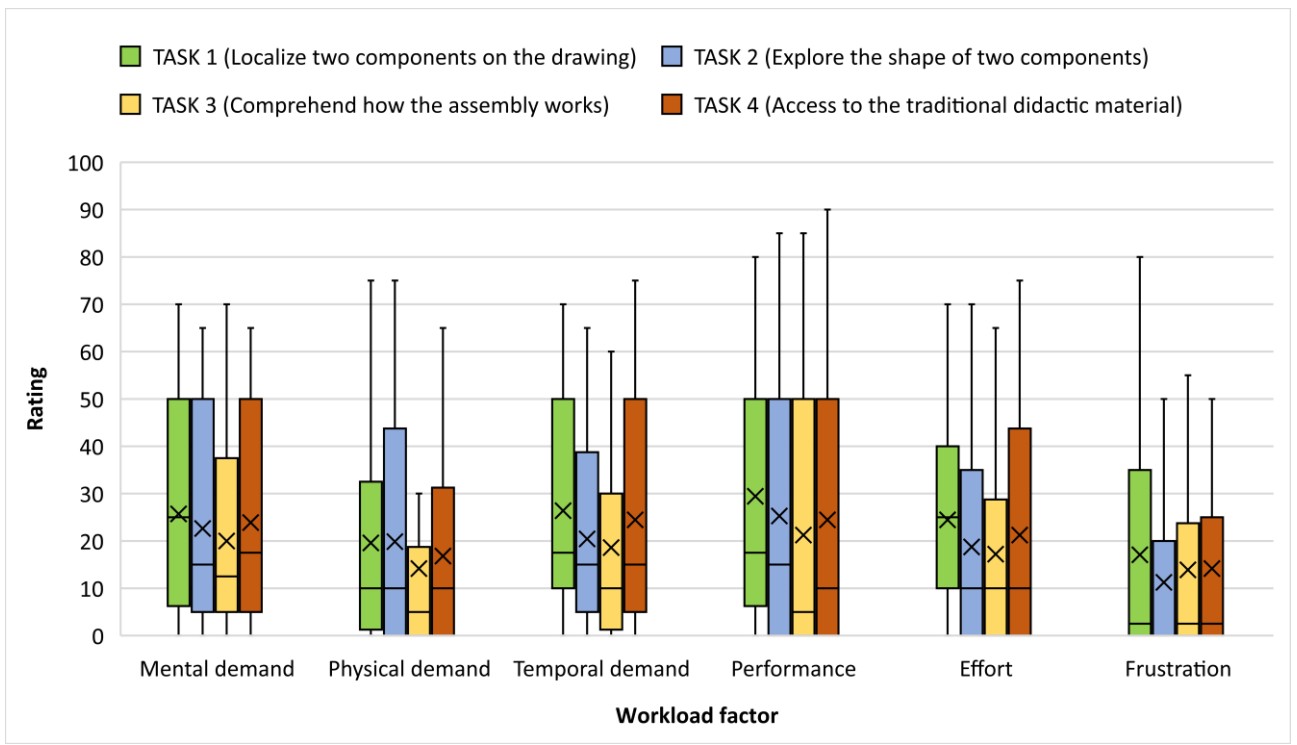

**Figure 13.** Rating of the six workload factors of NASA TLX according to learning tasks. The "X" indicates the mean workload factors score for each task.

For all the four learning tasks, the mean weighted NASA TLX score was in the range (10–29), corresponding to a medium cognitive load [43], thus respecting the SR-017.

## 5. Discussion

The results of this work allow us to answer our research questions:

1. "Which is the usability of the designed MR application? How is it affected by students' familiarity with MR and DE tools?"
2. "Which is the cognitive load associated with the learning tasks in the MR application?"

The mean SUS score is 84.7, which allows us to define the usability of our application as excellent [42]. Furthermore, it is not influenced by students' familiarity with MR and DE tools. This result allows us to say that favorable judgments are not due to the "wow-effect" that could bias novel users' ratings. Then, in future MR applications for other STEM disciplines, the same sample of students could evaluate usability and cognitive load. However, students familiar with MR and DE tools know the main issues and the potentialities of these tools; for this reason, they could provide more useful feedback.

The mean weighted NASA TLX score for all four learning tasks accomplished through the proposed MR application is between 10 and 29, which allows us to define the cognitive load associated with the MR application as medium [43].

The results of both user studies allow us to say that the application can be optimized for future usage, considering students' feedback in the first user study. However, it can be introduced effectively during a class at this current stage, and learning skills acquired through the application can be tested.

With the MR application proposed in this work, we provided a way to solve the main issues about the use of MR for STEM didactics, reported in Section 2:

- Teachers are scarcely motivated to the use of MR in STEM education;
- A high effort is required for the development of virtual content;
- MR applications have distracting graphics that further do not assist students in carrying out learning activities, whereas increasing cognitive load;

- Integration of traditional didactic material into MR applications is needed;
- MR applications should stimulate senses other than sight.

We considered users not only the students but also the teachers. This consideration allowed us to consider teachers' needs and involve them in the design and development of the application. As pointed out by [7,8], teachers' involvement is crucial for students' perception of the usefulness of this tool. Recent research [44] confirms the importance of taking teachers at the center of the design. The author proposed a methodology that, similarly to our approach, is based on interviews with students, teachers, and parents to define user needs for AR learning experience in primary schools. The authors aimed at finding the right balance between traditional teaching and self-learning supported by AR technology, but without providing neither implementation nor a validation of their design.

The effort required for the authoring of virtual content is one of the limiting factors for using MR as a learning tool, as revealed by [7]. In our study, the initial development of the application framework still requires knowledge of 3D development tools and AR software development kits. However, using a scalable framework and an essential interface with only virtual content needed for learning allows at least to reduce the authoring effort to create new learning activities for the target discipline. Our case study revealed that a skilled student could successfully do this task in no more than four working days. This relevant result could further motivate teachers to effectively use these tools in schools and universities. They can commit this task to motivated students, making possible a virtuous circle where students design learning tools for other students.

Another novel aspect in our study compared to others in similar applications in the literature [15,45] is the use of basic graphics. We proposed an intuitive interface using, where possible, graphics to which students are accustomed. For example, the selection menu of the machine components is structured as the Bill of Material present in the 2D drawing of the machine. Furthermore, we used basic shading for the CAD models with one uniform color for every component instead of a more realistic shading. This choice has two benefits: on the one hand, it allows students to distinguish exactly one component from the others, which is one of the learning aims. On the other hand, it allows a lower effort in the authoring of the application, using only the minimal visual assets required to convey the information, as suggested in [46]. The results of our work say that the use of basic graphics contributes to reducing the cognitive load in performing the learning tasks, without affecting usability, even if many students proposed an enrichment of the application: e.g., use of a realistic rendering of CAD models, visualization of the assembly animation in a real environment, displaying of technical details. These suggestions are beneficial and can be discussed for future applications. However, we must consider that previous works in this field [19,27] revealed that rich graphics and an excessive amount of simultaneous information might distract students from their primary learning task. Thus, using AR and desktop VR instead of immersive VR could be better because the virtual elements displayed are only those needed to convey information for learning. The approach of avoiding distracting graphics is also used in recent similar research presented by [16], where authors presented an AR application to help students manipulate electrical circuits by applying Kirchhoff's laws. The authors used similar tools and design strategies but without providing a methodology. Additionally, the motivation for the work is the same. They explored the possibility of using AR for laboratory activities in situations where people cannot meet in large numbers, such as in emergencies related to SARS-CoV-2. They found that the attitude towards using the AR application does not depend on the training carried out by a teacher in a guided laboratory class (face-to-face). This result is comparable to that of our work, where we found that the usability of the MR application does not depend on students' familiarity with MR tools. A similar result can be obtained by basic graphics in the interface that allow self-learning.

Another novel approach followed in our application is related to the application's aim. It helps in acquiring or integrating knowledge usually acquired through laboratory activities. Therefore, it does not aim to comprehend all the theoretical arguments presented

in books or teacher booklets. However, we designed an additional section in our application, where the main theoretical concepts are also presented, reusing the same contents available in the teacher booklets. Then, our application integrates but does not replace the traditional didactic material, thus following the advice that comes from the literature [17,26]. The user evaluation results confirm that the aim of the application is met. Only 15.2% of students agree or strongly agree that they can learn without using traditional didactic material with the MR application. On the other hand, it should be noted that this application cannot replace all the laboratory activities. In total, 45.5% of students agree or strongly agree that they would prefer taking a closer look at real machine components as carried out in the laboratory activities. This significant result justifies a gradual use of MR for STEM education, confirming what was found by [47]. They argue that with the continuous progress in computer graphics and VR interfaces in the future, virtual laboratories may eventually reduce the need for real-world laboratories. Currently, they are useful tools in emergencies when laboratories are not accessible to students. However, when laboratory activities can be conducted in the presence, the use of these applications is still justified. They are proper complementary learning tools, offering many advantages for students. They allow a potentially unlimited number of case studies that cannot be explored in a laboratory. Every student can learn at his/her own pace and practice with dangerous and expensive tools without risks.

Regarding the stimulation of senses other than sight, we only limited ourselves to sounds during the real machine simulation. This limitation is mainly due to the target device chosen: the student's own smartphone or tablet. An immersive VR experience would also allow the use of artificial smells or haptic feedback for students. However, in the context of DE, this scenario would not be sustainable for students due to the high costs of the hardware needed.

A potential limitation of this work is that the discipline found as a case study has a high added value with MR. There is an excellent learning benefit in the association between real, i.e., the drawing, and what is virtual, i.e., the CAD model of a component. However, the design of the application is still valid for other STEM disciplines, even for such of them where there is a lower added value with MR. Only user studies (steps E and F) would reveal an advantage or not in the introduction of MR also for these disciplines. Another limitation of this work is that we only evaluated the usability of the developed application. The usability study results are enough to consider our application accepted by students for its introduction to the course. We planned to improve the application further in future works, considering students' feedback during this user study. We will then introduce the application in the engineering course, and we will perform a second user study to evaluate the learning effects of the MR application.

## 6. Conclusions and Future Works

In this work, we presented an MR application framework to support STEM DE laboratory lectures. The methodology used for the design of the MR application is based on the analysis of user needs for the definition of user requirements and system requirements. One of the relevant aspects of our research is that we considered users—not only the students but also the teachers. We involved them in the design and development of the application. We proposed an easily scalable framework for new learning activities for the same discipline, thus reducing the effort required to author new content. Another novel point of our approach with respect to other MR applications for STEM education is the use of simple and less distracting graphics. In this way, we present only the relevant information for their learning to students. Even if our application integrates the traditional didactic material, thus responding to one of the user needs, it has not the scope of replacing theory lectures. Its primary scope is to provide knowledge usually acquired during practice activities in laboratory lectures. The MR application was designed for a case study of an engineering course, developing an application to help students understand assembly drawings of complex machines. However, our design approach and the framework of our

application can also be used for other STEM disciplines. We evaluated the usability of our MR application, and we found that the positive results were not due to the wow-effect. We further evaluated the cognitive load associated with the learning tasks accomplished through the MR application, and it turned out to be medium. These results show that the designed application can be distributed to all students in the engineering course. In future works, we plan to improve the application further, considering the feedback given by students during this user study. Then, we will introduce the application in the engineering course, and we will perform a user study to evaluate the learning effects of the MR application.

**Supplementary Materials:** The following are available online at https://politecnicobari-my.sharepoint. com/:x:/g/personal/michele_gattullo_poliba_it/Eb53IDMblvtHn25glbd4T24Bc6u7TG7m475SdbGejID0 yA?e=sBCGLU, Table S1: SUS data.

**Author Contributions:** M.G.: Conceptualization, Methodology, Formal analysis, Investigation, Writing—Original Draft; E.L.: Methodology, Software, Visualization, Investigation, Writing—Original Draft; A.B.: Supervision, Resources, Funding acquisition, Writing—Review and Editing; A.E.: Software, Investigation, Validation, Writing—Review and Editing; M.F.: Supervision, Resources, Funding acquisition, Writing—Review and Editing; V.M.M.: Software, Investigation, Writing—Review and Editing; A.E.U.: Supervision, Resources, Funding acquisition, Writing—Review and Editing. All authors have read and agreed to the published version of the manuscript.

**Funding:** This work was supported by the Italian Ministry of Education, University and Research under the Program "Department of Excellence" Law 232/2016 (Grant No. CUP-D94I18000260001).

**Institutional Review Board Statement:** Not applicable.

**Informed Consent Statement:** Informed consent was obtained from all subjects involved in the study. Written informed consent has been obtained from the participant(s) to publish this paper.

**Data Availability Statement:** Data is contained within the article.

**Acknowledgments:** The authors would like to acknowledge all the students and teachers involved in this research study for their time and precious feedback.

**Conflicts of Interest:** The authors declare no conflict of interest.

## Appendix A. User Needs, User Requirements, and System Requirements

**Table A1.** User needs specific of the course of "Methods for Technical Representation," deriving from focus group interviews with teachers.

| ID | User Need | Source |
|---|---|---|
| UN-005 | Students need to learn how to identify machine components in an assembly drawing | Students, Teachers |
| UN-006 | Students need to learn how to associate the real shape of a machine component with its 2D representation | Students, Teachers |
| UN-007 | Students need to understand the functionality of a component within the machine | Students, Teachers |

**Table A2.** User needs specific of the course of "Methods for Technical Representation," deriving from a preliminary survey to students.

| ID | User Need | Source |
|---|---|---|
| UN-008 | Students need to have a learning tool that allows self-learning | Students |
| UN-009 | Students need to have a learning tool that stimulates the study of the subject | Students |
| UN-010 | Students need to have an intuitive learning tool | Students |
| UN-011 | Students need to have a learning tool that is deployable on their mobile device | Students |

**Table A3.** User requirements.

| ID | User Requirements | Description | Reference to User Needs |
|---|---|---|---|
| UR-001 | Essential and usable interface | The application will be easy to use, easy to develop, stimulating but without distracting elements | UN-001, UN-002, UN-009, UN-010 |
| UR-002 | Reduced cognitive load | The learning tasks must be accomplished without producing a high cognitive load for students | UN-001, UN-010 |
| UR-003 | Use of available resources | The application will use drawings traditionally used by the teachers of the course or available in the book. In this way, it will be easy to access the 3D CAD models used to generate those drawings | UN-002, UN-003 |
| UR-004 | Additional information for a component | The application will contain a section with additional information for a specific type of component, such as those available in the traditional didactic material | UN-003, UN-008 |
| UR-005 | A tool to identify a component in the drawing | The application will have a tool to associate the name of a component with its location on the drawing | UN-005 |
| UR-006 | A tool to associate the real shape of a machine component with its 2D representation | The application will have a tool to display the 3D model of a component directly on the drawing | UN-006 |
| UR-007 | Tools to understand the functionality of a component | The application will have animations and sounds that simulate the real machine | UN-007 |
| UR-008 | Application deployed for mobile devices | The application should work well on both smartphones and tablets | UN-011 |
| UR-009 | Easy authoring of new contents | The framework of the application should allow the deployment of a new learning activity with a low authoring effort | UN-002 |

**Table A4.** System requirements.

| ID | System Requirements | Priority | Description | Reference to User Requirements |
|---|---|---|---|---|
| SR-001 | Help text for students not expert with MR | W | In the GUI, there will be a text instruction that suggests how to frame the drawing correctly | UR-001 |
| SR-002 | Automatic hiding of contents | S | Only information related to foreground objects are displayed | UR-002 |
| SR-003 | Drawings used for natural feature tracking in the AR section | M | The drawings will be used as markers; they will be generated from CAD models already modeled by master's degree students | UR-003, UR-005, UR-006 |
| SR-004 | Desktop VR section of the app | M | Navigable 3D models of single components with labels to explore details | UR-004 |
| SR-005 | Section of the app with additional information at the request | M | Students can ask for information available on traditional didactic material | UR-004 |
| SR-006 | Interactive Bill of Material (BOM) | M | In the GUI of the app, there will be an interactive BOM table where users can select the component to highlight | UR-001, UR-005 |
| SR-007 | "Highlight" button | M | Button to highlight a component on the drawing | UR-005 |
| SR-008 | "Balloon" button | M | Button to display a component positioning number on the drawing | UR-005 |
| SR-009 | "3D model" button | M | Button to display a 3D model of a component on the drawing | UR-006 |
| SR-010 | "ALL" button | M | Button to display the 3D model of the entire machine | UR-005, UR-006, UR-007 |
| SR-011 | "Animation" button | M | Button to play/pause animations | UR-007 |
| SR-012 | "Compare" button | C | Button to display a half-section 3D model of the entire machine with labels to compare with the 2D Section on the drawing | UR-005 |
| SR-013 | "List" button | C | Button to access the Section of the app with additional information on components | UR-004 |
| SR-014 | Adaptive design of the interface | S | The interface will adapt to the resolution of the display | UR-008 |
| SR-015 | Scalability of the framework | S | The creation of new learning activities in the project is fulfilled by making a copy of an existing one, replacing the contents | UR-009 |
| SR-016 | Good usability | M | The SUS score must be higher than 75 | UR-001 |
| SR-017 | Low cognitive load | M | The NASA TLX index must be lower than 30 for every learning task | UR-002 |

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
