# Peer review of "Design of a Mixed Reality Application for STEM Distance Education Laboratories"

_computers, doi:10.3390/computers11040050_

Round 1

Reviewer 1 Report

The virtual/augmented lab issues are not new in the STEM distance education laboratories context, but there is still resistance to introducing new practices in this domain.

The paper shows how to answer the two following questions: “Is the usability of the designed application good enough for its introduction in a real course?” , and  “Is usability influenced by students’ familiarity with Mixed Reality and distance education tools?”.

Authors provided MR applications to students with different familiarities levels with MR and DE tools. Results show that usability of the application is fine and that it is not linked to familiarity with Mixed Reality and distance education tools.

At this second step, the paper, its structuration, content, illustration have been upgraded substantially, and the presentation have also been improved. Thus, the paper can been accepted as that.

Author Response

We thank the Reviewer for taking the time to leave us the feedback. We are delighted to hear that the Reviewer appreciated the improvements of our work.

Reviewer 2 Report

 This paper proposed a Mixed Reality (MR) application to support laboratory lectures in STEM distance education. Some development methods and  usability,  cognitive load are presented for its practical use. 1. From the paper, what is the methodology? There are no title of section for presentation. 2. It can seen that the vuforia is used for drawing identification. But the tool is not mentioned in the paper. 3.There are so many pictures. But not all pictures are necessary and some important pictures are missing, e.g. the picture of the main function (maybe explored model). The pictures should show the learning content of the students. 4.In the time evaluation of the system, the model should be given to show the complexity of the objects, which will influence the using time. 5. The literatures should be given for SUS,QUIS, SGUS in 3.3.1. 6. For the TLX evaluation, only the results are given. It will be better to show the original score in proper style.  

Reviewer 3 Report

First, congratulations on writing such a valuable article. I would also like to thank you for your dedication to writing the article.

However, I propose to improve the article. Detailed comments are included in the .pdf file.

Round 2

Reviewer 2 Report

I suggest the acception of the paper.

Reviewer 3 Report

I think the authors have significantly improved the manuscript and it can be accepted in it’s current version.